# Patient Information and Informed Consent for Research in the Elderly: Lessons Learned from a Randomized Controlled Trial

**DOI:** 10.3390/healthcare10061036

**Published:** 2022-06-02

**Authors:** Maria López-Parra, Francesc Zamora-Carmona, Mònica Sianes-Gallén, Esmeralda López-González, Dolors Gil-Rey, Helena Costa-Ventura, Miriam Borrás-Sánchez, Gemma Rayo-Posadas, Marta Arizu-Puigvert, Roser Vives-Vilagut

**Affiliations:** 1Parc Taulí Hospital Universitari, Institut d’Investigació i Innovació Parc Taulí I3PT, Universitat Autònoma de Barcelona, Parc Taulí 1, Sabadell, 08206 Barcelona, Spain; fzamora@tauli.cat (F.Z.-C.); msianes@tauli.cat (M.S.-G.); elopezg@tauli.cat (E.L.-G.); mdgil@tauli.cat (D.G.-R.); ecosta@tauli.cat (H.C.-V.); grayo@tauli.cat (G.R.-P.); marizu@tauli.cat (M.A.-P.); 2Institut Català de la Salut, North Metropolitan, 08924 Barcelona, Spain; mborrassa.mn.ics@gencat.cat; 3Departament de Farmacologia, de Terapèutica i de Toxicologia, Universitat Autònoma de Barcelona, Cerdanyola del Vallès, 08193 Barcelona, Spain; roser.vives@uab.cat

**Keywords:** aged, informed consent form, nurses, randomized controlled trial

## Abstract

The informed consent (IC) of subjects participating in experimental studies is the mainstay to comply with the ethical principle of autonomy to ensure that the participation is voluntary. This experience was performed within the context of a single-center randomized clinical trial in elective prosthetic surgery. Obtaining IC in clinical trials is not without difficulties, and especially in the case of vulnerable populations it can be very challenging. This work aimed to identify the difficulties during the IC process for a clinical trial in subjects older than 65 years old and quantify and describe the use of IC in front of a witness. Methods: This is a mixed methodology study with a qualitative part (focus group with 4 nurses involved in the inclusion of subjects) and a quantitative part describing the characteristics of patients who signed IC forms. Results: The main difficulties identified are related to comprehension, sensory impairments, education level, and time. IC in front of witnesses was used in 20 patients out of 508. Conclusions: The participation of subjects older than 65 years old in clinical trials requires an adaptation of the process. The use of IC in front of a witness should always be considered in studies including elderly subjects.

## 1. Introduction

The search and application of the best scientific evidence is an essential requirement for healthcare professionals. The continuous search for excellence guarantees that the safest and most effective treatments and cares are offered to the population.

The awareness that research is essential to improve evidence-based care has increased among nursing professionals. Different research methodologies contribute with different levels of evidence; however, it is acknowledged that clinical trials are the gold standard to demonstrate the cause–effect relationship between an intervention and a result [1].

When implementing experimental studies, ethical principles must go first. Since 1964, the Helsinki declaration of the World Medical Association has laid down the main ethical principles for research in humans, among them, the need for the informed consent (IC) of subjects participating in research, thus ensuring that the participation is voluntary [2]. For this reason, each potential subject must receive accurate and truthful information of all aspects of the trial with special stress on the anticipated potential risks of the study and the right to withdraw the consent at any time. Only after the investigator can ensure that the subject has understood the information, informed consent can be obtained. These ethical principles have been adopted by different national and international regulations [3].

The IC is considered the paradigm of the care relationship between the patient and health professionals to guarantee the bioethical principle of autonomy, and it can be defined as the express manifestation of a subject of the willingness to participate in the proposed activities, accepting freely and voluntarily the conditions, benefits, and risks that this involves [4].

In accordance with these principles, the standards of good clinical practice [3] specify that before any study procedures are initiated, the subject (or his/her legal representative) and the professional responsible for the information must sign and date the consent form [5]. It also contemplates situations in which the patient is not able to read or write, in which an impartial witness, who is present during the information process, testifies to the patient’s consent.

Obtaining IC for a clinical trial is not without difficulties, especially in the case of vulnerable populations: pregnant women, children, patients with psychiatric disorders, and the elderly, among others [3,6]. In these situations, comprehension problems, under-literacy of subjects and factors associated with senility can be very challenging. It has been described that the poorer the academic level of the potential participant, the less is the understanding of the key aspects of the research in which participation is offered [7]. Besides, the difficulty of understanding information increases in aged patients [8,9]. On the other hand, short, simple, and easy-to-read information ensures understanding [10]. 

Under the premise that advanced age does not have to be an exclusion criterion for participation in randomized clinical trials, it is necessary to take into account the barriers to obtaining informed consent for this population group.

The present investigation aimed to identify the difficulties in providing information, filling in, and signing IC forms that arise during the recruitment of subjects over 65 years of age. In this age group, the most frequent pathologies are osteoarticular degenerative diseases, and the most prevalent are those that affect the knee and hip joints, which may require an arthroplasty. This population is frequently affected by different geriatric syndromes, including different degrees of delirium, depression, and sensory deterioration (blindness and deafness), some of which directly impact not only their comprehension ability but also their literacy skills. This experience is part of the development of a randomized clinical trial of prosthetic surgery to compare 5 types of wound dressings. It was carried out in a reference university hospital that serves a reference population of 489,000 inhabitants [11] by a research team of nursing professionals. As mentioned, this work was conceived within the context of a single-center randomized clinical trial to compare 5 different wound dressings in elective prosthetic surgery. In this study, age was not limited per inclusion criteria; however, due to the characteristics of patients undergoing this type of surgery, it was to be expected that a considerable number of eligible patients would be over the age of 65. The recruitment started in April 2017 and ended in December 2019, with a target recruitment of 550 patients.

The process for including patients in this study starts during the routine outpatient pre-surgical visit performed by a nurse, which takes place approximately one week before the planned surgery day. This visit aims to check the health status of patients and that no changes in pharmacological treatments have occurred since the anesthesiology visit. During this visit, the patients receive detailed information on practical aspects of the procedure with a special focus on postoperative wound care and recovery.

Within the context of the trial, patients eligible to participate were identified during this visit (patients who are candidates for the fast-track procedure). The nurse informs the patients of the possibility to participate in the trial and provides comprehensive information to obtain informed consent from patients who agree to participate.

As per best practice research (PBC), risk-adjusted monitoring of the trial data was performed by an independent monitor. During the first monitoring visit, all informed consent forms of patients included were reviewed. In 27 out of the first 50 forms, the monitor detected that the investigator had assisted the patient to fill in some of the data of the form, such as the name of the patient, the name of the informing investigator or the date. A total of 23 of these patients (85%) were older than 65 years old. The reason argued by the investigators was that they intended to facilitate the process and that they were not aware that it was the patient who had to handwrite most of the information in the form. This finding led to this study. Thus, the monitor advised that whenever possible these fields should be filled in by the patients, and if this was feasible, an oral informed consent in front of a witness (ICW) had to be implemented. Bearing in mind these situations and the recommendation of the monitor, we contemplated two questions to begin this research: Which difficulties do we have when obtaining informed consent in subjects older than 65 years old? How often is it necessary to use the ICW?

The objectives of the present work were to identify the difficulties during the process of informing and obtaining IC to participate in a clinical trial, in subjects older than 65 years old and how to resolve them. Additional objectives were to quantify the frequency of use for the many occasions we used the ICW, to detail in which situations it has been used, and to describe the demographic characteristics of subjects that used ICW.

## 2. Materials and Methods

This was a mixed methodology study with a qualitative part (focus group) and a quantitative part with data obtained from a clinical trial in patients undergoing elective hip or knee replacement surgery.

The investigator team considered it of interest to conduct a qualitative investigation and decided to run a focus group with the four nurses who were mainly recruiting patients into the trial. The questions we worked on during the two focus group sessions were: “Which difficulties do you have when obtaining informed consent in subjects older than 65 years old and how do you resolve them?” The focus group aimed to share the experience and points of view of each nurse regarding the difficulties during the recruitment process and to identify conflicting points. Two investigators analyzed the answers following the method of the focus group, from the written transcription (literal note-taker) of the comments. Repeatedly reading the notes of the focus group, we assembled the topics, and we identified categories that coded and analyzed the contents to elaborate a final report [12].

In June 2017 the possibility of obtaining informed consent in front of a witness was implemented with the previous approval of the Ethics Committee, to overcome some difficulties observed during the process. 

At the end, a descriptive analysis of all informed consent forms obtained from June 2017 (when we introduced the ICW, from the subject number 50 on) up to December 2019 has been performed, when the recruitment of the clinical trial was finished.

The study variables were, for the qualitative part, the difficulties in informing and obtaining consent identified through the focus group and key strategies to resolve it. For the quantitative part, the number of informed consent forms obtained, the types of informed consent and the demographics characteristics of patients for the whole sample and the different types of informed consent (age, biological sex, type of surgery, number of ICW, and reasons for use). The results of this part are described as absolute and relative frequencies for categorical variables and as means and standard deviations for quantitative variables.

The clinical trial was previously approved by the ethics committee of university hospital in January 2017 (ID number: 2017/014) and is being performed according to GCP. The ICW amplification was presented and approved by the same committee as an amendment, and the subjects were informed. 

## 3. Results

From June 2017 to December 2019, 558 patients had given informed consent to participate in the clinical trial. Eventually, 558 patients were recruited, but the ICW was introduced from subject number 50 on, so 508 ICs of the recruited subjects were analyzed for this study. Of those, 348 were over 65 years old, 60.5% were female and 72.9% of them were undergoing knee surgery.

ICW was use in 20 patients (3.9%). The social demographic characteristics of these patients are described in Table 1. Two subjects were under 65 years old. Of the mentioned 20 subjects, 80% were female. The witness was usually the caregiver accompanying the person.

The main reason for using this informed consent was difficulty in writing that would make the filling out of the form too slow (12 cases), illiteracy (4 cases), and visual impairment (4 cases).

### Focus Group Results

From the focus group, the following categories were identified (Figure 1). Potential solutions to these difficulties were discussed:


*“Some elderly patients have difficulties understanding clinical language.”*


Concerning comprehension, the investigators said that they progressively learned how to adapt the language used to explain the study mainly to the elderly patients. During the information, the investigator shows the patient a sample of each of the wound dressings and explains randomization as a “lottery”. Some patients express their discomfort as they cannot choose the dressing.


*“Frequently, they show a preference for one of the wound dressings, thus, it is essential to remark that they will be randomly assigned, as a lottery.”*


The investigators observed that the provision of information and the process of obtaining of informed consent took longer with older patients, both to guarantee that the patient understands the key aspects of the study and for the signature of the form.

They also emphasized that the process may be more complicated in patients with deficient reading or writing due to sensory impairments (visual and hearing impairment) and fine psychomotor:


*“Sometimes they haven’t got their glasses and can’t read the documents… also patients with ‘essential tremor’ have difficulties signing informed consent.”*


Another difficulty was the low education level of some patients:


*“In such cases, we often observe that patients are suspicious, thus, we remark the kind of information that we want to obtain from the study (…), another difficulty that we face is the low education level of the patients, as sometimes it is difficult for them to read and understand the information sheet and the informed consent form they must sign.”*


Various subjects declined to read the informed consent information and form:


*“Some of them refuse to read the informed consent… They complain: “I’m in a hurry, I haven’t got the glasses, It’s too long, I will read at home…”*


At the time of decision-making, the investigators observed that the eldest group of patients had difficulties in making up their mind and tended to ask for advice from the accompanying relatives or the investigator concerning their decision:


*“…a frequent question when we ask patients if they have any doubt regarding the study is: what would you do?”*


Investigators stated that it is necessary to stress the patient’s autonomy, offering the patient the information sheet to be read during the visit. However, almost all patients preferred to read it at home:


*“This can be related to the lack of time of the patient or the accompanying person.”*


## 4. Discussion

The present study has been performed by using data of a clinical trial in patients with knee or hip arthroplasty. This condition is highly prevalent in the elderly population; in our sample, 68.5% of patients were older than 65 years. This is a good setting to analyze the process of information and signature of the IC form in this age range. The characteristics of the studied population are very similar to other investigations performed in orthopedic surgery [13,14,15]. After the ICW form was implemented, it was used only in 20 cases. We observed that some subjects didn’t want to use it if they could write, even though slowly. It is noteworthy that its use was more frequent in women. This observation might probably be related to a higher rate of illiteracy in elderly women, because they had fewer opportunities for schooling in childhood. They might also be in a hurry because they have to look after grandchildren or other family members.

The focus group identified the following categories, and solutions for improvement were proposed for improvement: Comprehension problems, visual and hearing impairment, deficient literacy skills, rush or lack of time, request for advice for decision making.

The main difficulties identified are in agreement with the ones reported by different authors [16,17,18].

This study intends to stress the relevance of effective communication and simplicity of information when recruiting patients older than 65 years in clinical trials.

Even though including elderly subjects may increase the time needed for the inclusion visit, it is deemed necessary to include this elderly population. Excluding this age group from clinical trials may lead to biased results and a lack of applicability of results to this population of patients, which on the other hand are the target population of several interventions [5,19].

One of the problems identified was the difficulty in comprehension. Explaining the methodology of a clinical trial to a patient may be difficult and one of the most challenging things is to explain the randomization. According to Weinfurt [17], up to 14% of patients could not reproduce correctly the information provided by the investigator. Another challenge is to determine the ability of the patient to understand the information [20]. In our study, patients with clear evidence of cognitive impairment or who were very doubtful were discarded straight away, according to our exclusion criteria.

To enhance comprehension, empower the patient and observe his/her autonomy, we explained thoroughly the different options to which the patient could be assigned (characteristics of the dressings), showing to them in situ the different dressings studied.

Some publications mention the difficulties in recruiting elderly patients in randomized clinical trials [17,18] and outline the difficulty in ensuring that the patients understand all aspects of the experiment, assuming the potential risks and benefits. This may require a longer time for the investigator who is recruiting, time that is not always available.

In our study, the impression of the lack of time is not only for the investigator (30 min assigned per patient) but due to the rush of the patient or the accompanying person. This may explain why they sign IC without properly reading the information sheet.

Despite the fact that it is recommended to allow sufficient time so that the subject can read thoroughly and understand the characteristics and the aims of the study [21], in our setting this is not always feasible as the visit agendas are tight, and there’s seldom the possibility to reschedule a second visit to sign de IC after the patient has read the information at home. In our case, the information and the signing of the IC must be done at the same visit; thus reading the information sheet at home before signing is not an option.

The assistance burden, the lack of time and, especially, the traditional asymmetry between the health personnel who follow an indicative paradigm and the patients who are historically passive are the main barriers that block good communication, interfering with the ethical aspects of the physician–patient relationship and consequently favoring a paternalistic behavior [22], thus jeopardizing the patients’ autonomy.

In elderly patients, a tendency to this paternalistic model has been observed, which may explain why patients asked for the opinion of the recruiter before making the decision.

The shared decision is a process by which the choice of medical actions is made together by the health professional and the patient, and it is nowadays a key consideration. However, despite the fact that it is changing, health care professionals, according to their will to do what’s best for their patients, often assume a paternalistic attitude when making decisions for them. In a considerable number of occasions, these actions are executed without the patient being adequately informed and without them being completely aware of the consequences of the medical process [23]. Consequently, this is translated also to the situations of clinical research.

Beyond any conceptual definition, the respect for the patient’s autonomy and the protection of their biological, psychical, and social integrity mostly rely on excellent communication and a good communication strategy begins by putting aside the paternalistic tone [22].

On the other hand, a qualitative study to identify in which way nurses could encourage autonomy (self-determination and free choice) in elderly patients has been published recently [24]. In this study, Jacobs et al. underline the importance of giving full information, highlighting the pros and cons of the decisions and the relevance of the participation and compromise, and explaining that the whole process should be individualized. All these considerations were already implemented in our clinical trial. The findings of the focus group support the importance of these strategies.

In the health care setting, it is essential to transition to a paradigm of more participative relationships, in which we treat patients instead of diseases and where diversity is considered.

Putting into practice a perspective based on the dialog will allow health care professionals to articulate a pluralist perspective [22].

It is common practice to obtain the IC of the patient without paying much attention to the information process. Patients should be persuaded neither by the investigators nor by the accompanying persons as it is a sign that the patient is not ready to give his/her IC. It must be regarded that there will always be a pool of patients who decline participation. According to Mallia, this would be a sign of correctness of the recruiting process, where the subject’s autonomy is observed [3].

As previously mentioned, the lack of literacy skills was also detected as one of the problems. In Spain, there’s a functional illiteracy rate of 1.7% among the population older than 16 years; in particular, this problem affects 399,600 people older than 70 years old [25]. This partly explains the difficulties encountered with patients of this age range and was one of the reasons why the first IC forms were not properly filled in, as some patients were able to write their names but writing the name of the investigator and the date was somehow difficult for them.

After the ICW was implemented, it has been used a few times, a fact that confirms that most patients can write their names and sign and to copy the investigator’s name and the date, as long as they are given the time needed according to their abilities. Patients giving their ICW were older than the rest of the subjects included in the study and most of them were women. This may be related to a higher illiteracy rate among women in our country. The main reason was that the patient had slow writing, a fact that may be related both to illiteracy and to visual and/or psychomotor difficulties.

One of the particularities of our work is that it was performed in the setting of a single-center study. This means that few investigators did the recruitment of a considerable number of patients, allowing the identification of specific problems, the proposal of solutions and ultimately a training of the investigator who has developed specific communication abilities. The expertise of the investigator team in informing and obtaining the informed consent of subjects for research guarantees that the information is transmitted correctly and that the autonomy of the patient is observed. It is not the usual situation that nurses are the leaders of a research project despite being potentially skilled in communication abilities with the patient in terms of language adequacy, detection of needs, and attention to families.

The difficulties identified may appear in other studies including patients aged 65 years or older and the solutions proposed could be extrapolated. Further investigations should focus on multicenter studies and in different sociocultural contexts, considering globalization.

In this case, the implementation of the ICW allowed the participation of a few subjects, ensuring their autonomy. This modality of IC may be underused despite the fact that it should always be considered in studies, including elderly patients, to facilitate the process.

## 5. Conclusions

The main difficulties identified during the process of informing and obtaining IC to participate in a clinical trial in subjects older than 65 years old are related to comprehension, sensory impairments, education level, and availability of time.

To resolve these difficulties, the research team used communication techniques, using plain and direct language, active listening, non-verbal communication, and empathy are key points when informing patients.

The number of ICW used was low—20 times—considering the total size of 508.

The main reason for using this informed consent was difficulty in writing that would make the filling out of the form too slow. To observe the autonomy principle, the participation of subjects older than 65 years in clinical trials requires an adaptation of the information given and availability of time.

Subjects that used ICW were mainly elderly and women. Thus, we must recommend that the possibility of using the ICW is always included, offered, and considered in studies including elderly subjects. It could also be considered in other vulnerable populations with physical impairments.

## Figures and Tables

**Figure 1 healthcare-10-01036-f001:**
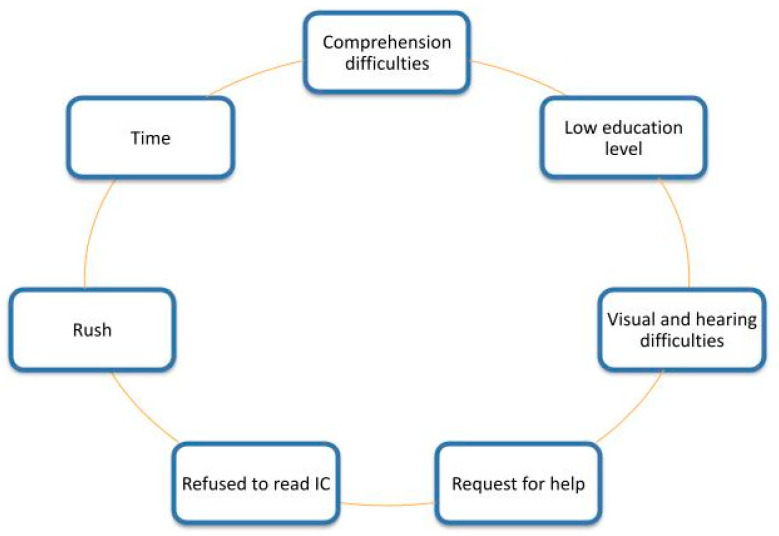
Topics related with informed consent in adults over 65 years.

**Table 1 healthcare-10-01036-t001:** Sample Description.

Variables	Total (*n* = 508)	>65 Years (*n* = 348)	Needed IC Witness (*n* = 20)
Age Mean (SD)	69.10	74.22	73.91
	(±9.68)	(±5.81)	(±13.69)
	(35.65–93.81)	(65.01–93.81)	(59.4–88.5)
>85 years *n* (%)	65	65	2
	(12.8%)	(18.7%)	(10%)
Female *n* (%)	308	214	16
	(60.5%)	(61.5%)	(80%)
Knee *n* (%)	371	278	16
	(72.9%)	(79.9%)	(80%)

Characteristics of patients giving informed consent.

## Data Availability

The datasets used and/or analyzed during the current study are available from the corresponding author on reasonable request.

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
