# Peer review of "Patient Information and Informed Consent for Research in the Elderly: Lessons Learned from a Randomized Controlled Trial"

_healthcare, 2022, doi:10.3390/healthcare10061036_

Round 1

Reviewer 1 Report

The topic of this study is important, and worthy of investigation. However, neither the methods nor the results were described adequately, making it difficult to assess the scientific soundness of the research, or whether the inferences and conclusions were reasonable.

Ultimately, it was not clear what the real focus and purpose of the study was. Initially it seemed like the study was focusing on patient comprehension of informed consent forms. However, the bulk of the study seemed to focus on the nurses perceptions' of hindrances to obtaining informed consent, which could include issues of comprehension, but also other things like time constraints, the impact of vision impairments, etc. The author needs to clarify the focus of the study, perhaps by including some research questions.

To be ready for publication, I would suggest the following areas need to be addressed:

The author has to describe more clearly the "study within a study" nature of this research. That is, the study of informed consent seems to take place within the context of a larger research study focused on wound dressings of elderly patients getting hip or knee replacements. To clarify the current study, the author needs to:

Explain what motivated this study

Indicate exactly how subjects were recruited (it sounds like a decision to launch the current study might have come after some irregularities were discovered in the consent forms of patients in the larger study. Is that true? If so, it needs to be described in more detail).

Explain the ethical parameters of the current study. The author notes that the study was reviewed by an ethics board, but it is unclear if they mean the current study or the larger one. The reader needs to know if the current study was reviewed. Also, were the elderly patients alerted to the current study? Or were their forms collected and analyzed without their knowledge?

More detail needs to be given about the nursing focus groups. What kinds of questions were asked? How was the data collected (note-taker, audio recording, etc.)? Provide more information about how the data was analyzed and the codes developed.

The presentation of the results is somewhat unclear, especially as regards the patients' consent forms. The author lists several different numbers-- they say they had a target of 550 forms, but at one point they talk about 27 out of 50 forms having irregularities, and elsewhere they say they had 262 forms. What was the final number and how was it obtained? How do the first 50 forms fit in? Finally, the only result given related to the forms is that 17 involved a witness. Nothing more is said about irregularities. Really, this finding doesn't tell us anything about whether any of the patients understood the information in the forms. I think the author needs to clarify what the purpose and take-aways of this part of the study is?

The focus group results also need more detail. The author lists 7 themes, but does not explain how they were developed. Further the author refers to two of the themes and provides some quotes to illustrate them, which is helpful. 

The discussion seems to draw on the findings of the focus group but because these findings were not described in detail it is impossible to know if the discussion and conclusions are valid.

Author Response

 We would like to thank you for your interest and your review of our manuscript. We consider all of your comments of great value and have been of great help to improve the manuscript. We hope that the new version incorporating the changes can be considered for publication.

Best regards,

Reviewer 2 Report

The manuscript entitled: “Patient information and informed consent for research in elderly: Lessons learned from a randomized controlled trial” aimed to identify the difficulties during the IC process for a clinical trial in subjects older than 65 years old and to quantify and describe the use of IC in front of a witness.

The manuscript is very interesting, however, some aspects still need to be improved, especially the novelty and scientific contribution.

The scientific gap and the novelty are still not clear. Is this paper innovative or just an application and analysis of the methodology in Spain? Please, clarify the paper's scientific contribution.

The objective should be clarified in the Introduction. There are only 3 lines to describe it. Or there are more objectives?

Data could be more detailed presented.

Please, revised references according to the journal’s rules.

Author Response

 We would like to thank you for your interest and your review of our manuscript. We consider all of your comments of great value and have been of great help to improve the manuscript. We hope that the new version incorporating the changes can be considered for publication. 

Best regards!

Round 2

Reviewer 1 Report

The authors adequately addressed my comments.

Reviewer 2 Report

Thank you for your revised version.

I think that the manuscript could be proceed further for publication